# Preprocessing Effects on Performance of Skin Lesion Saliency Segmentation

**DOI:** 10.3390/diagnostics12020344

**Published:** 2022-01-29

**Authors:** Seena Joseph, Oludayo O. Olugbara

**Affiliations:** ICT & Society Research Group, Luban Workshop, Durban University of Technology, Durban 4001, South Africa; seenaj@dut.ac.za

**Keywords:** histogram clustering, image preprocessing, melanoma diagnosis, morphological analysis, Otsu thresholding, saliency segmentation, skin lesion

## Abstract

Despite the recent advances in immune therapies, melanoma remains one of the deadliest and most difficult skin cancers to treat. Literature reports that multifarious driver oncogenes with tumor suppressor genes are responsible for melanoma progression and its complexity can be demonstrated by alterations in expression with signaling cascades. However, a further improvement in the therapeutic outcomes of the disease is highly anticipated with the aid of humanoid assistive technologies that are nowadays touted as a superlative alternative for the clinical diagnosis of diseases. The development of the projected technology-assistive diagnostics will be based on the innovations of medical imaging, artificial intelligence, and humanoid robots. Segmentation of skin lesions in dermoscopic images is an important requisite component of such a breakthrough innovation for an accurate melanoma diagnosis. However, most of the existing segmentation methods tend to perform poorly on dermoscopic images with undesirable heterogeneous properties. Novel image segmentation methods are aimed to address these undesirable heterogeneous properties of skin lesions with the help of image preprocessing methods. Nevertheless, these methods come with the extra cost of computational complexity and their performances are highly dependent on the preprocessing methods used to alleviate the deteriorating effects of the inherent artifacts. The overarching objective of this study is to investigate the effects of image preprocessing on the performance of a saliency segmentation method for skin lesions. The resulting method from the collaboration of color histogram clustering with Otsu thresholding is applied to demonstrate that preprocessing can be abolished in the saliency segmentation of skin lesions in dermoscopic images with heterogeneous properties. The color histogram clustering is used to automatically determine the initial clusters that represent homogenous regions in an input image. Subsequently, a saliency map is computed by agglutinating color contrast, contrast ratio, spatial feature, and central prior to efficiently detect regions of skin lesions in dermoscopic images. The final stage of the segmentation process is accomplished by applying Otsu thresholding followed by morphological analysis to obliterate the undesirable artifacts that may be present at the saliency detection stage. Extensive experiments were conducted on the available benchmarking datasets to validate the performance of the segmentation method. Experimental results generally indicate that it is passable to segment skin lesions in dermoscopic images without preprocessing because the applied segmentation method is ferociously competitive with each of the numerous leading supervised and unsupervised segmentation methods investigated in this study.

## 1. Introduction

Malignant melanoma is a deadly archetype of skin cancer diseases which is one of the primary causes of increased cancer mortality rates [1,2,3,4]. The early detection of melanoma in skin lesions is widely recommended to mitigate complications and high mortality rates caused by the disease [5,6]. The detection of skin lesions in dermoscopic images using the human eyes is an arduous process because of the visual similarity between benign and malignant skin lesions. Dermoscopy is a non-invasive system for the visual examination of a substructure of skin to assist the investigation of amelanotic lesions [7,8]. However, the diagnostic accuracy of dermoscopy is heavily dependent on the experience of a dermatologist and visual assessment is highly onerous, subjective, and non-productive because of the complex nature of dermoscopic images [4]. These intrinsic curbs can be mitigated with the help of a computerized dermoscopic analysis system that allows for fast and accurate decisions in detecting skin lesions [4,9,10]. Computer-assisted diagnosis is of paramount importance to increase the accuracy and efficiency of the diagnosis of skin lesions [11,12]. The automated systems for detecting skin lesions comprise three stages; lesion segmentation, feature extraction, and feature classification [4]. However, efficient segmentation of skin lesions is essential among these three stages because it helps to segregate skin lesions from the surrounding skins [7,13]. The success of the subsequent stages is heavily dependent on the preliminary segmentation output [4,5,6,10,14,15,16,17,18]. In addition, the segmentation process helps to identify local and global clinical features of the region of interest [10]. The accuracy of segmentation results of skin lesions is directly proportional to the accurate diagnosis of malignant melanoma.

The research trend in methods for segmenting skin lesions is generally focused on the improvement of segmentation accuracy and efficiency. Myriads of image segmentation methods have been proposed in the literature for skin lesions. They include thresholding [17,19], clustering [20,21,22,23], statistical region merging [8,10], saliency [19,24,25,26,27,28,29], and deep learning [2,30,31,32,33,34,35]. Although multifarious image segmentation methods exist in the literature, accurate segmentation of skin lesions is still a challenging open problem because of the heterogeneous properties of dermoscopic images. The asymmetric features such as the irregular border, uneven shape, and colors of skin lesions can cause difficulties in accurately segregating the regions of skin lesions from the background. Moreover, images with low contrast between skin lesions and normal skin, the presence of undesirable artifacts such as color charts, marker ink, ruler marks, air bubbles, vignette, noise, and inherent cutaneous artifacts such as hair and blood vessels can make the segmentation obligation more challenging [3,5,6,12,16,32,36,37]. The artificial correlation created by these undesirable artifacts could adversely affect the performance of segmentation methods for skin lesions. Most of the existing studies have applied different image preprocessing methods to mitigate the debilitating effects of undesirable artifacts in dermoscopic images. These preprocessing methods include contrast enhancement [5,7,19,20,37], illumination correction [3,6], and hair removal [3,9,12,26,34,36,37,38].

The preprocessing step is generally considered an effective way of achieving better segmentation results, but it compromises the processing speed. Besides, most of the existing segmentation methods are extremely dependent on the utilization of preprocessing methods, and related parameters to match the heterogeneous properties of input images can be hard to determine [30,34]. A generalized solitary method to handle the various types of undesirable artifacts is still an unresolved problem in the disciplines of image processing and computer vision. Hence, myriads of studies have relied heavily on different stages of preprocessing to address the countless types of undesirable artifacts and uneven color contrast of dermoscopic images [2,6,9,17,20,23,26,37,39,40,41]. The demand for the inclusion of multiple preprocessing phases can adversely affect the performance of a segmentation method while limiting its practical applications in real clinical settings. In addition, the generalizability of different segmentation methods across a wide gamut of images is highly restricted by the type of preprocessing steps employed.

The principal objective of this study was to investigate the effect of image preprocessing on the performance of a saliency segmentation method for skin lesions. The objective was accomplished by leveraging the collaboration of color histogram clustering [42] with Otsu thresholding [43] to realize the CHC-Otsu algorithm. The DullRazor artifact removal algorithm [44] and contrast limited adaptive histogram equalization (CLAHE) [45,46] are the two famous preprocessing methods utilized for experimentation in this study. It is interesting to report that both types of preprocessing methods have not significantly increased the segmentation performance of the CHC-Otsu algorithm. This outcome advocates the sufficiency of the CHC-Otsu algorithm to handle undesirable artifacts and color imbalance between the regions of skin lesions and non-skin lesions in dermoscopic images. The CHC-Otsu algorithm has been experimentally compared to numerous leading supervised and unsupervised segmentation methods to validate its performance. The experimental comparisons were based on publicly available dermoscopic datasets and standard performance evaluation metrics. The comparative results indicate that the CHC-Otsu algorithm can produce ferociously competitive results when compared to the leading methods investigated in this study. The core contributions of the work reported in this paper are threefold:The application of color histogram clustering with Otsu thresholding for automatic identification of the number of homogeneous clusters and efficacious segmentation of skin lesions in dermoscopic images.The investigation of the effect of image preprocessing on the performance of a saliency segmentation method that leverages the collaboration of the CHC algorithm with Otsu thresholding for skin lesions.The evaluation of the performance of the CHC-Otsu algorithm against numerous leading image segmentation algorithms through extensive experimentation.

The remainder of this paper is succinctly summarized as follows. Section 2 comprehensively highlights the related studies. Section 3 covers the materials and methods used for experimentation while Section 4 elucidates the experimental results. Section 5 discusses the experimental results with a brief concluding remark.

## 2. Related Studies

Related studies provide the literature evidence that performances of the existing segmentation methods for skin lesions are highly dependent on the effectiveness of the preprocessing phases employed. However, there is no solitary preprocessing method that can generally be applied to resolve the different types of undesirable artifacts inherent in dermoscopic images. Consequently, several studies have incorporated multiple stages of preprocessing to tackle the heterogeneous properties of dermoscopic images to improve segmentation results [2,8,9,17,23,26,32,33,37,39,40,41,47]. The dependency on manifold stages of preprocessing confines the generalizability of the existing segmentation methods and increases their computational complexities. The study reported in [30] highlighted the curbs of using preprocessing methods and proposed a supervised segmentation algorithm based on a fully convolutional neural network that is freed from preprocessing. The visual characteristics of dermoscopic images were inferred by an iterative learning process in multistage fully convolutional networks to improve segmentation results, but with the added cost of high computational complexity. The superpixel-based segmentation method in [48] eluded the preprocessing step, but performance results are highly dependent on the granularity of superpixels. The review of related studies is accomplished in more detail by considering preprocessing methods and the actual segmentation methods employed for the analysis of skin lesions.

### 2.1. Preprocessing Methods Used in Segmentation Processes of Skin Lesions

The accurate segmentation of skin lesions is often prejudiced by the heterogeneous properties inherent in dermoscopic images. These heterogeneous properties include the presence of undesirable artifacts such as hair, blood vessels, color charts, ruler marks, marker inks, vignettes, noise, uneven illumination, and specular highlights caused by the acquisition processes of dermoscopic images. Most of the existing segmentation methods are highly dependent on various levels of preprocessing phases to circumvent the effects of undesired artifacts that could compromise the accurate segmentation of skin lesions. The occlusion resulting from undesirable artifacts can significantly hamper the accurate segmentation of skin lesions in dermoscopic images [34]. This challenge has led to the development of numerous artifact removal methods for occlusion in dermoscopic images. The artifact removal methods are based on thresholding [8,9,26], morphology [2,37], filtering [9], and DullRazor [12,17,28,36,38,39,47,48].

Similarly, image enhancement preprocessing methods are widely applied to correct the non-uniform illumination and low contrast nature of dermoscopic images. These enhancement methods are based on contrast adjustment [9,17], filtering [8,17,26,40], adaptive histogram equalization [37,49], and contrast limited adaptive histogram equalization (CLAHE) [40,41,50,51]. The CLAHE is widely recognized as the best method among the prevailing enhancement methods for preprocessing of medical images [40]. In addition, literature has shown evidence of preprocessing stages based on histogram [33], mean subtraction [31], deep learning [34], multiscale decomposition [21], adaptive gamma correction [23], Z-score transformation [52], and Frangi Vesselness filter [41]. The artifact removal and image enhancement algorithms are generally executed before the actual segmentation and postprocessing methods are applied to suppress the leftover noise. Table 1 summarizes the different preprocessing methods used by the related studies for the segmentation of skin lesions.

### 2.2. Methods Used for Segmentation of Skin Lesions

Approaches for the segmentation of skin lesions spanning different algorithmic methods were proposed over the years. The methods can be categorized primarily into supervised and unsupervised segmentation approaches. The supervised segmentation methods use a priori knowledge of the ground truth of a large training dataset of images, while the unsupervised methods are generally trained online during the segmentation process. The supervised segmentation methods focus on training, learning, and extraction of hierarchical image features from large image datasets using supervised machine learning methods such as support vector machines (SVMs), and convolutional neural networks (CNNs) [26,30,31,32,33,34,35]. The CNN-based deep learning methods have recently gained popularity for the segmentation of dermoscopic images [30,31,32,33,34,35]. The supervised segmentation methods exhibit high-performance results compared to the unsupervised complements, but their computational complexity is the main bottleneck. Moreover, they demand a high volume of training data with many parameters tuning for outstanding performance on the segmentation of skin lesions. However, obtaining high-quality annotated medical imaging data is a difficult and time-consuming obligation [25,28,53]. These requirements present a huge constraint for eclectic applications of supervised deep learning segmentation methods to support medical image analysis.

The approach of unsupervised segmentation mainly uses unsupervised machine learning methods such as expectation maximization, spectral clustering, and K-means clustering methods. They use data clustering algorithms to rapidly obtain results by performing fewer calculations in a combined model that separates the regions of interest from the background regions. Due to their elegance and efficiency, a wide gamut of unsupervised segmentation methods has been developed to date and applied for the analysis of skin lesions. These methods include thresholding [17,19], clustering [20,21,22,23], region merging [8,10,39], and saliency [19,24,25,27,28,29,38]. More recently, a privacy-preserving segmentation method based on the partially homomorphic permutation ordered binary (POB) number system was proposed for the segmentation of encrypted skin lesions [53]. Thresholding methods use primitive image features such as color or texture to create an intensity histogram for separating skin lesions from background regions. Even though thresholding is widely accepted because of its simplicity, it can only produce satisfactory results with bimodal images that present high contrast between the lesion and surrounding regions, but it is limited by the intensity distribution of the skin lesions [24,30,38]. Region merging methods generally apply to merge rules to iteratively discover and merge adjacent similar pixels from the seeded pixels. However, they often suffer from poor performance on images with heterogeneous properties, complex textures, and variegated colors [24]. Moreover, the leakage issue in the process of region growing through the weak boundary is another drawback of region merging methods [54]. Color cue is an important feature commonly used in clustering methods for skin disease identification [55] and several color-based clustering methods were proposed over the years for segmenting skin lesions [20,21,22,23]. However, the automatic identification of the optimum number of clusters in an image is a major challenge experienced by clustering methods. This is because the parameters of the initial clusters and cluster centroids have a non-trivial role in the segmentation results [18,56]. Dermoscopic images are intrinsically featured with focal regions of interest that reflect high contrast discrimination. The features have been exploited by saliency methods to achieve a significant advancement in the segmentation of skin lesions. This is because the features can rapidly help locate salient objects by efficiently analyzing the image surroundings [25,26,27,28,29].

Regardless of the numerous types of segmentation methods developed for skin lesions, efficiency and effectiveness are prejudiced by the presence of heterogeneous properties inherent in dermoscopic images. The segmentation method proposed in [6] focuses on eliminating the numerous types of artifacts innate in dermoscopic images. The authors used continuous-time wavelet transformation with a neighborhood-based region filling algorithm for hair detection and hair inpainting with adaptive sigmoidal function for illumination correction. The method proposed in [20] used average filtering for artifact removal and applied contrast enhancement to distinguish the boundaries of skin lesions. The study in [37] used morphological operations to remove hair artifacts and histogram equalization was applied for image enhancement. The method introduced in [26] used the hair-removing algorithm [57] to eliminate the effect of hair and homomorphic filtering [58] was used for illumination correction. The unsupervised SDI+ [9] method for segmentation of skin lesions employed preprocessing in several stages as proposed in [9]. The vignettes or dark corners are first eliminated in images by thresholding, then intensity with saturation features of the Hue saturation value (HSV) color model were exploited to address specular highlights, and bottom hat filtering was finally applied to remove hair artifact [9]. Table 2 summarizes the different methods used by the related studies for the segmentation of skin lesions on benchmarking datasets of international skin imaging collaboration 2018 (ISIC 2018), Pedro Hispano hospital (PH2), and human against machine 10,000 (HAM10000) with numerous numbers of images used for evaluation. Small-sized datasets such as PH2 have been widely used by many studies, while the huge HAM10000 dataset is only used in the current study.

The comprehensive review of the related studies has exposed the literature hiatus in the following summarization. The related studies underlined the dearth of a generalized solitary preprocessing method that can handle the numerous undesirable artifacts inherent in color images. The application of unsupervised image segmentation methods that incorporate preprocessing methods to improve performance results is pervasive for the analysis of skin lesions. There have been research efforts to abolish image preprocessing in supervised image segmentation, but work is continuing to achieve significant success. The usage trend in the removal of undesirable artifacts and image enhancement has shown DullRazor and histogram equalization-based methods to be widely used for removing undesirable artifacts and enhancing dermoscopic images, respectively.

## 3. Materials and Methods

The materials for this study include the experimental dermoscopic image datasets, algorithm implementation tools, and performance evaluation metrics. The primary method of this study is the CHC-Otsu algorithm, which leverages the collaboration of the CHC algorithm with Otsu thresholding. The preprocessing effect on the performance of the CHC-Otsu saliency segmentation method for skin lesions was investigated using a paired *t*-test, which is considered an adequate device for judging the significant difference between the means of two distributions [61,62]. Thus, the two-tailed paired *t*-test was used to estimate the statistical significance of the means of preprocessed and non-preprocessed performance distributions of the CHC-Otsu algorithm. The *p*-value less than 0.05 (*p* < 0.05) is considered to declare a hypothesis statistically significant or not. The statement formulated for the statistical analysis of the performance of the CHC-Otsu algorithm is explicated as follows. The performance of the CHC-Otsu algorithm for skin lesions has significantly increased by incorporating a preprocessing stage in the segmentation method at a 5% significance level. The null hypothesis (Ho) and alternate hypothesis (Ha) are formed based on this statement as follows.

**Ho:** *There is no statistically significant improvement in the performance of the CHC-Otsu algorithm in segmenting skin lesions by incorporating a preprocessing stage*.

**Ha:** *There is a statistically significant improvement in the performance of the CHC-Otsu algorithm in segmenting skin lesions by incorporating a preprocessing stage*.

### 3.1. Materials

The experimental datasets for this study are the publicly available PH2, ISIC 2018, and HAM10000. PH2 is a dataset of manual segmentation and identification of 200 dermoscopic images of melanocytic lesions with their corresponding ground truths performed by expert dermatologists [63]. ISIC 2018 is the largest public dermoscopic image dataset of the ISIC 2018 challenge with 2594 images and their ground truths for the analysis of skin lesions to detect melanoma [64]. HAM10000 is a training dataset of 10,015 dermoscopic images recently released for training deep learning methods through the ISIC archive [65]. The dataset is widely used for the classification of skin lesions, but literature shows no wider evidence of the performance of the images for segmentation [66,67,68,69]. The dataset has been used in this study to evaluate the segmentation results of skin lesions to observe the performance of the CHC-Otsu algorithm on a huge dataset. The implementation and evaluation of the CHC-Otsu algorithm were carried out using MATLAB (2019a, The MathWorks, Inc., Natick, MA, USA) on a computer with an Intel(R) Core (TM) i7-8650U CPU @ 1.90GHz 2.11 GHz and 8 GB RAM.

The evaluation metrics of accuracy, sensitivity, specificity, dice similarity, and running time are commonly used for evaluating the performance of a segmentation method for skin lesions [10,12,23,30,35,37,52,59]. Sensitivity is the amount of the correctly detected pixels of skin lesions while specificity is the ratio of the correctly segmented non-lesion pixels [10,70,71]. The dice similarity measures the association between the segmentation output and ground truth [72]. The running time of an algorithm for a specific input is directly dependent on the number of operations that will be executed in proportion to the size of the input.

The DullRazor [44] and CLAHE [45] are two devices used in this study to determine whether preprocessing based on artifact removal and image enhancement, respectively, have effects on the performance of a saliency segmentation method for skin lesions. DullRazor is a digital skin hair shaver that is widely used in many studies as a preprocessing device developed for removing dark hair from images, and clean dermoscopic images for further processing [12,19,34,36,38,39,48]. CLAHE is an effective and excellent preprocessing device broadly applied for contrast enhancement of natural and medical images [40,45,46]. It uses the local contextual region of an image to compute a pixel value instead of considering the entire image region.

### 3.2. Methods

The CHC-Otsu algorithm is a saliency segmentation method used to investigate the effect of preprocessing on its performance for the analysis of skin lesions. The algorithm is an integration of the CHC algorithm [42] with the Otsu thresholding algorithm [43] for saliency segmentation of skin lesions. Saliency segmentation methods were inspired by their ability to retrieve the most conspicuous objects from the background information in a manner reminiscent of the human visual system by observing the local or global visual rarities such as color, intensity, contrast, and brightness [73,74,75,76]. It ideally induces a multidimensional color image into a grayscale image that is naturally amenable to Otsu thresholding. This ability provides a source of inspiration for the unification of the two algorithms for the saliency segmentation of skin lesions in dermoscopic images. The CHC algorithm presents a simple, but efficacious procedure for saliency segmentation of objects in color images according to the following four essential steps: color image quantization, regional feature extraction, saliency map computation, and saliency map postprocessing.

Color quantization is a widely used process of reducing the number of distinct colors in an image by merging the less dominant colors into dominant ones with a resulting quantized image that has similar visual appeal to the original image. The CHC algorithm uses the ‘imquantize’ built-in color quantization function in MATLAB (2019a, The MathWorks, Inc., Natick, MA, USA) to effectively acquire the desired number of dominant colors of the input red, green, and blue (RGB) color image at level 8. This realizes a maximum number of 512 colors that corresponds to a maximum of 512 possible homogeneous regions in a color image. Thus, the color quantization process uses dominant and significant colors to renew the input RGB color image and reduce the computational complexity associated with the processing of color images [16,77,78,79].

The regional feature extraction is accomplished using the color histogram of the quantized input RGB color image to create 8 to 512 homogeneous color clusters depending on the color quantization result. Homogenous regions can employ richer feature representation for saliency detection than individual pixels of a color image [80,81]. The optimum determination of clusters using the color histogram is widely accepted by researchers. The histogram defines the frequency distribution of color image data to symbolize the count of pixels in the image [18]. A global color histogram with 8 × 8 × 8 bins is computed from the quantized RGB image for the automatic creation of homogeneous clusters for saliency computation. The four features of color contrast, contrast ratio, spatial feature, and center prior are extracted for each cluster. The color contrast of a cluster is determined by finding the color difference of the cluster in the normalized L*a*b* color model to all other clusters. A cluster shows high saliency when it has a high contrast to the nearby clusters than the distant clusters [82]. The contrast ratio is used to emphasize the high color contrast difference between the maximum and minimum brightness of clusters [42]. The spatial feature is the spatial correlation of a cluster to all other image clusters. The center prior is the distance of each cluster to the image center to emphasize a low weight for the cluster framed near the image boundary [80,83,84,85,86].

The regional saliency map is computed by agglutinating the extracted four features in the previous stage. The integration of all these features has enhanced the segmentation of skin lesions and made the CHC algorithm robust against the low-contrast regions, random positioning of skin lesions, and undesirable artifacts in an image. The cluster-level saliency is used to compute the final saliency score of each pixel as follows. Since the pixels belonging to the same cluster have the same saliency, the saliency value of each pixel is assigned by the saliency value of the respective cluster. Saliency map postprocessing is the last stage of the CHC algorithm because it is customary in image processing to exclude noise or leftovers after the final segmentation process. The CHC algorithm implements the grayscale morphological operations as a postprocessing step to achieve a smooth and accurate segmentation result. Morphological operations are applied in image segmentation methods to eliminate isolated pixels by closing and filling operations [9,12,17,19,20,22,26,28,36,60,87].

The MATLAB (2019a, The MathWorks, Inc., Natick, MA, USA) implementation of the Otsu thresholding is used in this study to complement the operation of the CHC algorithm. The Otsu method performs non-parametric, unsupervised, and automatic global thresholding in the segmentation process. The optimal threshold is chosen by a discriminant criterion to maximize the separability of the resultant clusters in a grayscale image. The procedure utilizes the zeroth-order and first-order cumulative moments of a grayscale histogram to determine an optimum threshold. The pixels of the input grayscale image are dichotomized into 2 clusters at a particular scale, then an optimum threshold is determined to create a Silhouette image. The inherent merit of Otsu thresholding is that the geometric characteristics of an image object do not affect its performance because it performs thresholding on image intensity [88].

The CHC-Otsu algorithm presents five steps: input image quantization, regional feature extraction, cluster saliency computation, Silhouette Otsu thresholding, and binary morphological analysis. Binary morphological operations of fill and dilation with a disk-shaped structural element of radius 3 are applied to the Silhouette map created by Otsu thresholding of the grayscale saliency map generated by the CHC algorithm. The image regions with an isolated area smaller than 30% are removed because of the fundamental assumption that skin lesions are typically the largest objects in a dermoscopic image. Algorithm 1 gives a full algorithmic description of the essential steps of the CHC-Otsu algorithm for segmenting skin lesions. The algorithm is generic and is applicable in other image segmentation assignments.

The important variables used in the algorithm are Index, Palette, and Cluster. The Index variable is of dimensions M by N, and it stores the index value between 1 to 512 associated with the quantized input image. The parameters M and N are, respectively, the height and width of the input image. Palette of dimension 512 by 5 stores the average L*a*b* color features, average spatial coordinates, and count of homogeneous pixels in a region of the input image. Cluster is a variable of dimensions K by 9 that stores the non-zero elements in the Palette and regional properties that characterized a cluster, where K lies in the closed range [8, 512]. Step 2 implements the process of input image quantization. The regional feature extraction process is implemented in steps 3 to 24. The process covers the transformation of the input RGB image to the L*a*b* color image normalized to values in [0, 1]. The empty clusters resulting from fewer colors detected are removed in step 25. The computation of cluster saliency score begins from step 26 to step 50 while the clusters are rescaled to [0, 1] in step 50. The task of assigning a saliency score to each image pixel is accomplished in steps 51 to 55. The Otsu Silhouette thresholding and binary morphological analysis, respectively, are executed in steps 56 and 57. The MATLAB (2019a, The MathWorks, Inc., Natick, MA, USA) norm function used in steps 30 and 44 implements the Euclidean distance. This algorithmic description of the CHC-Otsu Algorithm 1 is based on the mathematical formulation of the CHC Algorithm [42].
**Algorithm 1.** CHC-Otsu
Input: M × N × 3 RGB color image (Input), distance scaling parameter (*n*) such that 0.1 ≤ *n* ≤ 1.0Output: *M* × *N* Silhouette image (Output), number of clusters automatically detected (K) % Constant parametersNIC = 3; % number of image componentsIQL = 8; % image quantization levelNCD = IQL^3; % maximum number of desired colorsLCL = 1; % lab color lightLCA = 2; % lab color ALCB = 3; % lab color BPXC = 4; % pixel x-coordinatePYC = 5; % pixel y-coordinatePDC = 6; % pixel distance to image centerCPC = 7; % cluster pixel countCPL = 8; % cluster pixel labelCCC = 8; % cluster color contrastCSS = 9; % cluster saliency score 1.  K = 0;2.  Index = ImageQuantization(Input, IQL);3.  Input = rgb2lab(Input);4.  Input = rescaleImage(Input, [0,1]);5.  for x = 1 to M do6.     for y = 1 to N do7.      Palette(index(x, y), CPC) = Palette(index(x, y),CPC) + 1;8.      Palette(index(x, y), LCL) = Palette(index(x, y), LCL) + Input(x, y), LCL);9.      Palette(index(x, y), LCA) = Palette(index(x, y), LCA) + Input(x, y), LCA);10.      Palette(index(x, y), LCB) = Palette(index(x, y), LCB) + Input(x, y), LCB);11.      Palette(index(x, y), PXC) = Palette(index(x, y), PXC) + x;12.      Palette(index(x, y), PYC) = Palette(index(x, y), PYC) + y;13.     end for14.  end for15.  for z = 1 to NCD16.     if Palette(z, CPC) > 0 17.      K = K+1;18.      Palette(z, CPL) = K;19.      Palette(z, PXC) = Palette(z, PXC)/M;20.      Palette(z, PYC) = Palette(z, PYC)/N;21.      Palette(z, 1:PYC) = Palette(z,1:PYC)/Palette(z, CPC);22.      Cluster(K, 1:CPC) = Palette(z, 1:CPC);23.     end if24.  end for25.  Cluster(K + 1:NCD, :) = [ ];26.  Wr = Cluster(:, CPC)/(M*N);27.  for x = 1 to K28.      Cluster(x, CCC) = 0;29.      for y = 1 to K30.       Cluster(x,CCC) = Cluster(x,CCC) + Wr(y)*norm(Cluster(x,1:NIC)-Cluster(y, 1:NIC));31.      end for32.  end for33.  for x = 1 to M34.   for y = 1 to N35.      Cluster(Palette(Index(x,y),CPL),PDC) = Cluster(Palette(Index(x,y),CPL),PDC) + (x/M-0.5)^2 + (y/N-0.5)^2;36.   end for37.  end for38.  for z = 1 to K39.      Cluster(z, PDC) = Cluster(z, PDC)/(*n* * *n* *Cluster(z, CPC));40.  end for41.  for x = 1 to K42.      Cluster(x, CSS) = 0;43.      for y = 1 to K44.       Ds = norm(Cluster(x, PXC:PYC)-Cluster(y, PXC:PYC));45.       Phixy = (Cluster(x, CCC) + 0.05)/(Cluster(y, CCC) + 0.05);46.       Cluster(x, CSS) = Cluster(x, CSS) + Wr(y)* Phixy*exp(-Ds);47.      end for48.      Cluster(x,CSS) = exp(-Cluster(x,PDC))*(Wr(x)*Cluster(x, CCC)+ Cluster(x, CSS));49.  end for50.  Cluster(:, CSS) = rescale(Cluster(:, CSS));51.  for x = 1 to M52.    for y = 1 to N53.       Input(x, y, LCL) = Cluster(Palette(Index(x, y), CPL), CSS);54.    end for55.  end for56.  Output = OtsuThresholding(Input(:, :, LCL));57.  Output = BinaryMorphology(Output);end Algorithm


## 4. Experimental Results

This section focuses on the experimental results obtained by evaluating the effects of DullRazor and CLAHE preprocessing methods on the performance of the CHC-Otsu algorithm. In addition, the overall performance comparison of the algorithm against the leading segmentation algorithms is evaluated for skin lesions. The presentation of the experimental results is structured as preprocessing effects by visualization, preprocessing effects by statistical testing, runtime analysis of preprocessing effects, and performance evaluation of skin lesion segmentation. The heterogeneous properties identified in the dermoscopic images are listed in Table 3.

### 4.1. Preprocessing Effects by Visualization

The simplest way to demonstrate the performance of a segmentation method is by visual inspection. Dermoscopic images with undesirable heterogeneous properties are orthodoxly used to visually demonstrate preprocessing effects on the performance of a segmentation algorithm. The segmentation results achieved by the selected dermoscopic images using the CHC-Otsu algorithm with and without the application of preprocessing methods are illustrated in Figure 1, Figure 2 and Figure 3 for each dataset, respectively. Since the images from the PH2 dataset do not sufficiently cover numerous undesirable heterogeneous properties (Table 3), only the available image categories are included in Figure 1. The inclusion of DullRazor and CLAHE preprocessing devices for artifact removal and contrast enhancement has roughly produced good results. Nevertheless, the results have demonstrated the ability of the CHC-Otsu algorithm to remove undesirable artifacts and adequately highlight skin lesions without the need for preprocessing.

The ISIC 2018 is a complex dataset with images of undesirable heterogeneous properties as shown in Figure 2. The adequacy of the CHC-Otsu algorithm to handle these heterogeneous properties can be observed in Figure 2. It can be perceptibly observed that segmentation outputs of images with serial numbers 10 and 12 are adversely affected by contrast enhancement preprocessing.

The inclusion of contrast enhancement preprocessing sharply waned the segmentation performance on images with the serial numbers of 3, 9, 7, 10, 11, and 12, as shown in Figure 3. The ability of the CHC-Otsu algorithm to accurately segment skin lesions regardless of the heterogeneous properties is detectably obvious from the results shown in Figure 1, Figure 2 and Figure 3.

### 4.2. Preprocessing Effects by Statistical Testing

Statistical performance evaluation measures of accuracy and dice similarity are employed to test the effects of preprocessing on saliency segmentation of skin lesions. The paired t-test statistic is used to determine the statistically significant evidence of the difference between the means of the non-preprocessed and preprocessed segmentation results illustrated in Table 4, Table 5 and Table 6. Pairs 1 and 3, respectively, represent the accuracy and Dice scores for the segmentation performances without preprocessing and with artifact removal preprocessing. Pairs 2 and 4 represent the accuracy and Dice scores, respectively, obtained for the segmentation performance without preprocessing and with image enhancement preprocessing. The mean value of accuracy is significantly higher without the artifact removal preprocessing for the PH2 dataset. However, the use of image enhancement preprocessing for the PH2 dataset increases the accuracy significantly with a p-value less than 0.05. In terms of the Dice similarity, the CHC-Otsu algorithm without the application of DullRazor preprocessing recorded a higher value, but the increment is not statistically significant. In divergence, the inclusion of image enhancement preprocessing is to contribute a statistically significant difference in the accuracy and Dice scores with a p-value of 0.000 < 0.05 to accept the alternate hypothesis.

The results in Table 5 obtained for the ISIC 2018 dataset indicate that the inclusion of artifact removal or image enhancement preprocessing did not contribute to a statistically significant difference in the segmentation results for the ISIC 2018 dataset. The increment in results obtained is not at a statistically significant level even though the accuracy and Dice scores for without preprocessing are higher than those obtained with artifact removal preprocessing. However, the CHC-Otsu algorithm without the inclusion of image enhancement preprocessing is seen to significantly improve the segmentation quality in terms of the Dice score.

The segmentation results in Table 6 for the HAM10000 dataset have clearly shown the ability of the CHC-Otsu algorithm without preprocessing support to accurately segment skin lesions regardless of the heterogeneous properties of the dermoscopic images. Significant increments in accuracy and Dice scores were achieved by the CHC-Otsu algorithm without the inclusion of the artifact removal preprocessing. The average Dice score is significantly increased without the application of image enhancement preprocessing.

### 4.3. Runtime Analysis of Preprocessing Effects

The computational running times of the CHC-Otsu algorithm on the selected datasets with and without preprocessing are reported in Figure 4. The computation time per image in each dataset shows that preprocessing increases the time complexity of the segmentation algorithm as projected. The running times can be seen to increase to 0.50 and 0.54 s, respectively using the DullRazor and CLAHE processing methods on the PH2 dataset. The computation time per image in the HAM10000 dataset is comparatively lower than for the other datasets because the dimension of each image in the HAM10000 dataset is 256 × 256, which is in disparity to the dimension of 400 × 300 for images in the PH2 and ISCI 2018 datasets. Furthermore, the average time per method across the datasets is depicted in Figure 5 to show that preprocessing using the CLAHE method recorded the worst average running time. 

### 4.4. Performance Evaluation of Skin Lesion Segmentation Results

The segmentation results for skin lesions were compared against the leading supervised and unsupervised methods based on the widely used performance measures of accuracy, sensitivity, specificity, and dice similarity to demonstrate the merit of the CHC-Otsu algorithm. The results from Table 7 illustrate that the CHC-Otsu algorithm recorded the highest specificity value. The result is highly competitive with that of the YOLO supervised deep learning method that recorded the highest accuracy score of 0.93 while the CHC-Otsu algorithm achieved an accuracy of 0.92. The high specificity score of 0.98 is an indication of the ability of the CHC-Otsu algorithm to accurately predict the non-lesion pixels. The CHC-Otsu algorithm with the algorithm described in [60] recorded the second-best Dice score of 0.89, which is highly closed to the highest Dice score of 0.90 [10]. This result shows the effectiveness of the CHC-Otsu algorithm in differentiating a larger amount of skin lesion pixels from the non-lesion pixels. The SPCA recorded the lowest sensitivity score of 73%, regardless of the 95% specificity score recorded.

Table 8 shows the comparative results of this study with the highest specificity score of 0.99. The supervised method [34] and Attention ResU-Net [35] achieved the highest accuracy score of 0.93 and Dice score of 0.87, respectively. The highest sensitivity score of 0.87 was achieved in [34] and SDI+ [9]. Even though SPCA recorded a specificity score above 90%, it did relatively present the worst sensitivity score of 0.59. These results demonstrate the competency of the CHC-Otsu algorithm because it did record the second-highest accuracy score of 0.92 and the best specificity score of 0.99.

This study did incorporate a huge dataset of HAM10000 for the analysis of skin lesion segmentation results (Table 9) further to the performance evaluation of the results using the PH2 and ISIC 2018 datasets that have been widely used by the numerous existing skin lesion segmentation methods. Since we do not have access to the computer source codes of the other comparative methods, the CHC-Otsu algorithm was compared with SPCA and SDI+ on the HAM10000 dataset. The CHC-Otsu algorithm recorded the highest accuracy score of 0.91 and specificity of 0.99. The Dice score of 0.82 recorded by the CHC-Otsu algorithm is very close to the highest Dice score of 0.82 recorded by SDI+ [9]. Similar to PH2 and ISIC 2018 datasets, the SDI+ recorded the best sensitivity score of 0.88. The scores recorded by the CHC-Otsu algorithm have shown the superiority of the algorithm when compared to other leading methods on this huge set of dermoscopic images.

## 5. Discussions and Conclusions

Detailed discussions and concluding remarks are explicated in this section to dispassionately interpret the significance of the study results relative to the findings of the comparative methods and to synthesize the important message of the paper.

### 5.1. Discussions

The delineation of skin lesions is an important prerequisite for melanoma diagnosis using a computer-aided diagnostic system. The results of this study have demonstrated the capability of the CHC-Otsu algorithm to accurately segment skin lesions in dermoscopic images of varying undesirable properties. The robustness of the results obtained is a consequence of the unique integration of saliency features of color contrast, contrast ratio, spatial feature, and center prior in the CHC algorithm [42]. It is worth noting that the CHC-Otsu algorithm successfully addresses the inherent complexities of dermoscopy images without the inclusion of preprocessing.

The comparison of the results computed by the CHC-Otsu algorithm with and without preprocessing has revealed a statistically insignificant difference in employing a preprocessing method to prepare dermoscopic images for skin lesion segmentation. The investigation of preprocessing effects has shown that skin lesions surrounded by undesirable artifacts such as hair, color chart, ruler marks, and marker inks are successfully corrected with the inclusion of the DullRazor preprocessing. However, improved results were obtained by the CHC-Otsu algorithm without the inclusion of artifact removal preprocessing as shown in Figure 1, Figure 2 and Figure 3. It is important to observe that the inclusion of the CLAHE image enhancement preprocessing sharply deteriorated the segmentation results of images represented by serial numbers 10 and 12 in Figure 2 and images with serial numbers 3, 7, 9–12 in Figure 3.

The results of the t-test statistical analysis illustrated in Table 4, Table 5 and Table 6 further support the qualitative results shown in Figure 1, Figure 2 and Figure 3. The inclusion of artifact removal preprocessing as shown in Table 4, Table 5 and Table 6 did not contribute to a statistically significant increase in the segmentation accuracy and Dice similarity. The CHC-Otsu algorithm without preprocessing achieved the highest performance scores across datasets than with the inclusion of the DullRazor preprocessing. The CHC-Otsu algorithm significantly produced higher segmentation results without preprocessing of images from the HAM10000 dataset. However, the inclusion of image enhancement preprocessing showed some statistically significant effects of segmentation accuracy on the PH2 and HAM10000 datasets with a p-value of 0.000 < 0.05. These results have led to the inference that image enhancement preprocessing improves skin lesion segmentation results. However, the effect of the image enhancement preprocessing was not statistically significant using images from the complex ISIC 2018 dataset. In divergence, the CHC-Otsu algorithm without preprocessing statistically produced significantly higher Dice scores than the segmentation with image enhancement preprocessing across datasets except for the PH2 dataset. However, compared to the ISIC 2018 and HAM10000 datasets, PH2 is a small dataset that does not present a sufficient volume of images with various heterogeneous properties. This assertion emphasizes the suitability of the CHC-Otsu algorithm to handle heterogeneous properties of dermoscopic images.

The investigation of preprocessing effects has shown that the CHC-Otsu algorithm is capable of effectively segmenting skin lesions, regardless of the presence of undesirable artifacts such as ruler marks, hair, color charts, ink marks, vignette, noise, a fuzzy border, and multiple shades of colors without preprocessing. This is in disparity with the findings of studies, which stated that preprocessing is needed for efficient and robust analysis of skin lesions in dermoscopic images [17,21,34]. A plethora of studies have showed the huge impact of preprocessing to significantly increase the skin lesion segmentation results [9,21,23,34,40,47,69,87,89,90]. Conversely, the results of this study support the findings of [91] that the preprocessing effect is dependent on segmentation and postprocessing methods employed. An appropriate segmentation method can elude the extra demand of preprocessing and mitigate its computational complexity. This avowal has been proven in this study, which demonstrated that seamless application of color histogram clustering and a holistic mechanism integrating color contrast, contrast ratio, and spatial features with center prior have highly contributed to the accurate segmentation of skin lesions in dermoscopic images. The exclusion of preprocessing in segmentation algorithms can significantly contribute to extenuating the intrinsic computational complexity.

The results obtained by ASLM [37] on images from the PH2 dataset are relatively not promising when compared to the results computed by the CHC-Otsu algorithm regardless of preprocessing. The YOLO supervised segmentation method [12] recorded the highest segmentation accuracy on the PH2 images. However, this deep learning method depends heavily on preprocessing and the availability of a huge training dataset to obtain higher confidence in the deep learning architecture. Hence, the main constraints of supervised methods are the inherent computational complexity and heavy reliance on high-quality annotated datasets. The SPCA supervised learning method based on superpixel and cellular automata [36] recorded the least sensitivity score. Moreover, SPCA demonstrates a comparatively low performance against the rest of the supervised learning methods listed in Table 7. The highest sensitivity value was achieved with the SDI+ method [9] and it is important to observe that high sensitivity is insufficient to quantify the real performance because a 100% sensitivity score could be achieved by identifying every image pixel as a skin lesion region. The method reported in [48] attained reasonably good accuracy and sensitivity results on the PH2 dataset without preprocessing because of the ability to develop additional structural data from superpixels. However, its dependency on single-value superpixel granularity may lead to producing undesirable segmentation results as superpixel-based methods are highly bounded by the optimum selection of superpixel granularity. It is important to note that this method did not manage to precisely detect the non-lesion pixels because the specificity score achieved by this method was only 0.89. In divergence, the CHC-Otsu algorithm is well developed to automatically determine the optimum region granularity and it achieves highly commendable segmentation results.

The deep learning methods of the R2AU-Net [34,35] obtained the highest accuracy score of 93% on the ISIC 2018 dataset. The performance of the CHC-Otsu algorithm on this dataset is laudable because it achieved the second-highest accuracy score of 92%, which is very close to the accuracy level obtained by the supervised methods. The Dice similarity performance of all the comparative methods can be seen to decline on images from the ISIC 2018 dataset when compared to the PH2 dataset. It reveals the complexity of images that occurred in this dataset and the difficulty experienced by all the comparative methods to obtain segmentation results that are closer to the ideal ground truth. The deep learning methods [34] and R2AU-Net [35] recorded the highest Dice score of 0.87 but the CHC-Otsu algorithm recorded a score of 0.81. The SPCA showed the worst performance on this dataset except for the specificity score. The performances of the SPCA and SDI+ on the ISIC 2018 data are relatively lower than their performances on the PH2 dataset because the ISCI 2018 dataset is presented with images of different undesirable heterogenous properties as listed in Table 3. This perception clearly emphasizes that preprocessing using the DullRazor and multiple preprocessing used by the SPCA and SDI+ algorithms, respectively, have not adequately assisted with enhancement of the segmentation results. The preprocessing employed by SDI+ [9] was focused on addressing only a few undesirable artifacts such as hair, specular lights, dark areas, and marker ink. The presence of other artifacts such as color calibration charts, and dark corners that can confound the performance of segmentation results appeared to be a challenging case for SDI+ [9] regardless of the inclusion of preprocessing. The authors of SDI+ observed the importance of having a preprocessing step to address the different types of undesirable artifacts.

The deep learning methods, as illustrated in Table 8, always show promising performance with the additional cost of computing resources. The demand for the huge training dataset and higher confidence in the deep learning architecture is directly proportional. consequently, there is always a tradeoff between computational complexity and efficiency performance of deep learning methods. The accuracy and generalization of deep learning methods relied on the availability of a large training dataset. However, regardless of the presence of complex images in the ISIC 2018 dataset, the CHC-Otsu algorithm shows surprisingly outstanding performance on this dataset with a simple but effective segmentation approach. The CHC-Otsu algorithm did outperform other algorithms on images from the HAM10000 dataset, and the segmentation results obtained have proved the robustness of the algorithm for segmenting skin lesions in dermoscopic images from a huge dataset.

Finally, the running times reported in Figure 4 and Figure 5 show the computational efficiency of the CHC-Otsu algorithm. One of the apparent disadvantages of employing a preprocessing phase is the demand for extra computation time, which is obvious from the results reported. The computation time of the CHC-Otsu algorithm increased with the incorporation of the preprocessing. The segmentation algorithms that use preprocessing phases to accelerate accuracy are adversely affected by the extra computational time with more debilitating consequences for methods that use multi-level preprocessing. The segmentation results are generally impeded by the undesirable heterogeneous properties inherent in the dermoscopic images. Mostly, these heterogeneous properties are addressed with the help of an appropriate selection of parameter signatures and preprocessing methods by the majority of the existing skin lesion segmentation methods. Due to the dependency on preprocessing methods, the generalized application of these methods is highly restricted [30]. Most of the comparative methods investigated in this study, except the ones proposed in [10,60] are highly dependent on the selection of preprocessing and parameters tuning. The method reported in [60] is freed from preprocessing, but its dependency on single-scale superpixel granularity may limit its ability to segment skin lesions in dermoscopic images with different features such as large areas, irregular boundaries, and undesirable artifacts. The results of this study have demonstrated the ability of the CHC_Otsu algorithm to accurately segment skin lesions without the constraint of preprocessing. In general, the findings of this study have illustrated that if a segmentation algorithm is adequate to address the complexities of dermoscopy images, the additional requirement on preprocessing can be excluded.

### 5.2. Conclusions

The preprocessing effects on the performance of skin lesion saliency segmentation have been meticulously investigated in this study using the CHC-Otsu algorithm which is robust, efficient, and independent of preprocessing. The algorithm has produced ferociously competitive segmentation results by taking the advantage of the collaboration of color histogram clustering with Otsu thresholding. Experimental results have shown the ability of the algorithm to effectively handle the various types of undesirable artifacts inherent in dermoscopic images. The performance of the algorithm has been extensively evaluated using the well-recognized publicly available datasets of the PH2, ISIC 2018, and HAM10000. However, the CHC-Otsu algorithm needs further improvement regardless of the overall performance, especially concerning sensitivity enrichment. The future work will focus on the inclusion of additional features such as color texture to improve the quality of the segmentation output and advance dermoscopic image analysis.

## Figures and Tables

**Figure 1 diagnostics-12-00344-f001:**
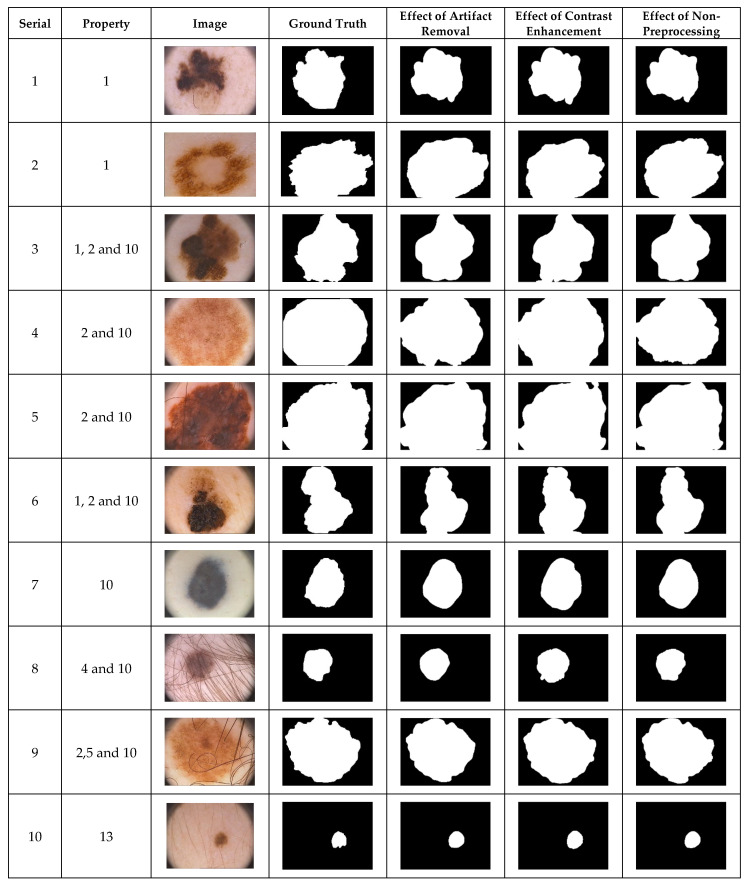
The visual effects of preprocessing on the PH2 dataset.

**Figure 2 diagnostics-12-00344-f002:**
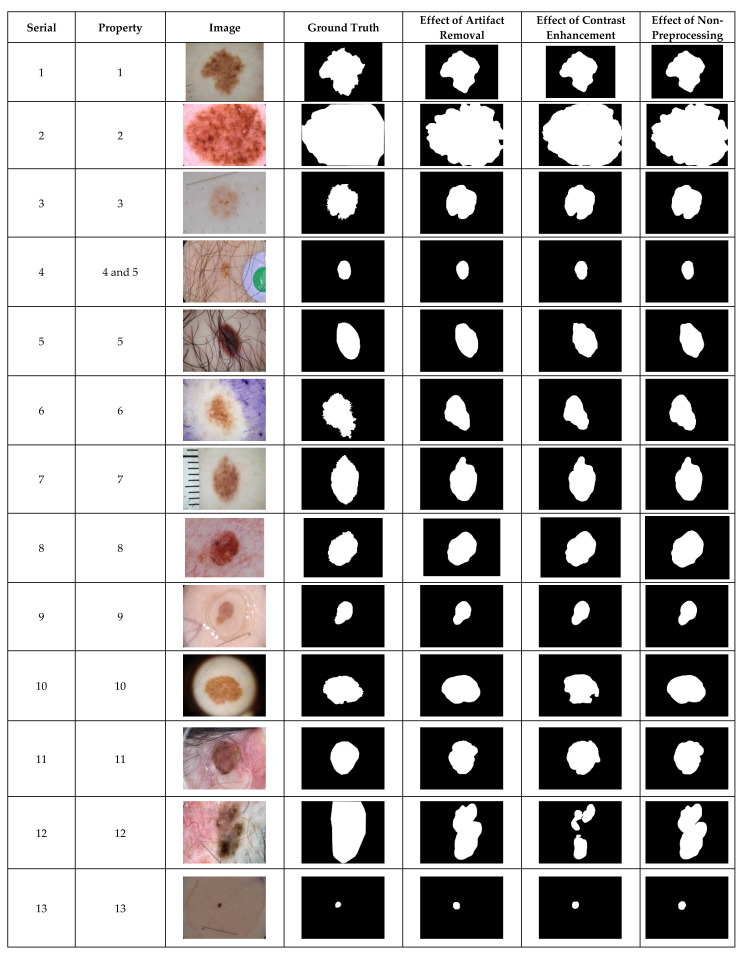
The visual effects of preprocessing on the ISIC 2018 dataset.

**Figure 3 diagnostics-12-00344-f003:**
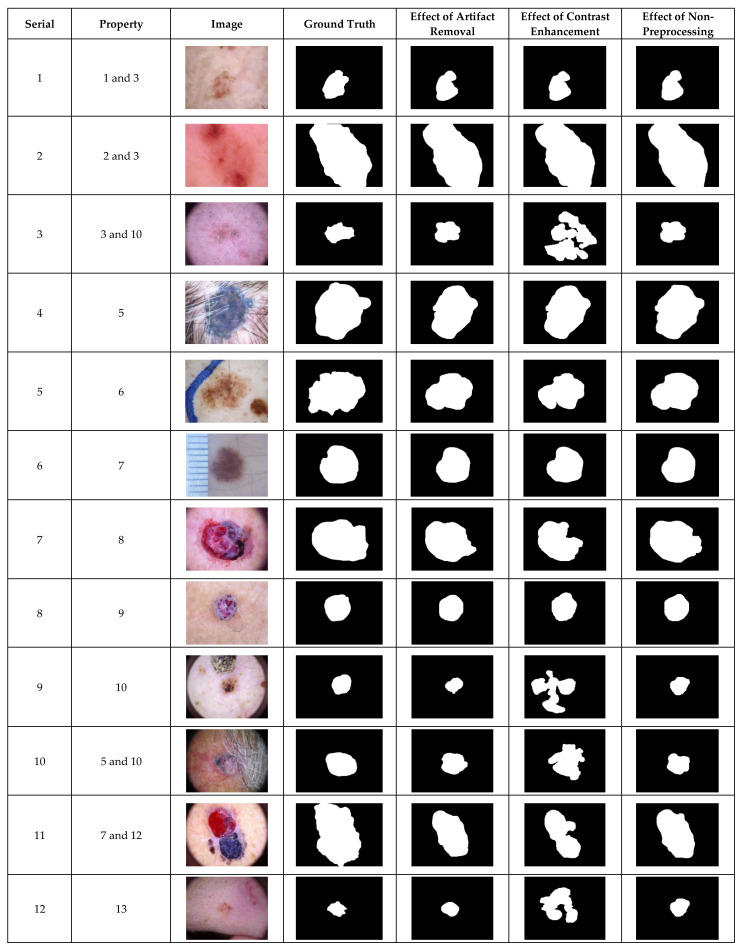
The visual effects of preprocessing on the HAM10000 dataset.

**Figure 4 diagnostics-12-00344-f004:**
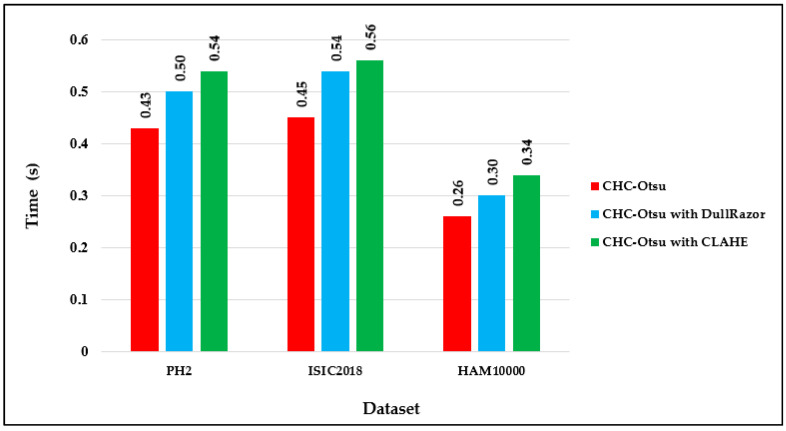
Computation time per image dataset.

**Figure 5 diagnostics-12-00344-f005:**
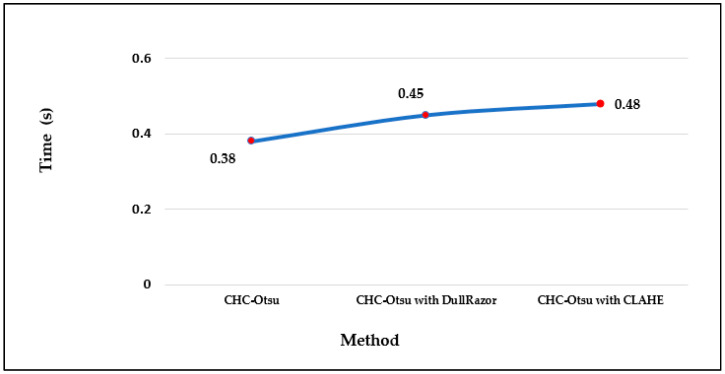
The average running time per method.

**Table 1 diagnostics-12-00344-t001:** Image preprocessing methods used for segmentation of skin lesions.

Study	Method
[2]	Artifact removal using morphological operations and image enhancement with unsharp filtering.
[8]	Artifact removal using thresholding and image enhancement with a median filter.
[9]	Artifact removal using the bottom-hat filter, dark corner removal with thresholding, and color enhancement by the intensity with saturation features of the HSV color model.
[12]	Artifact removal using DullRazor.
[17]	Artifact removal using DullRazor and image enhancement by noise filtering with intensity adjustment.
[19]	Artifact removal using improved DullRazor and image enhancement with top–bottom filtering, contrast stretching, and log transformation.
[20]	Artifact removal using averaging filter and image enhancement with contrast enhancement.
[21]	Artifact removal using multiscale decomposition.
[22]	Image enhancement using contrast enhancement method.
[23]	Artifact removal using a fast line detector and image enhancement with gamma correction.
[25]	Artifact removal using DullRazor.
[26]	Artifact removal using threshold decomposition and image enhancement for illumination correction with homomorphic filtering.
[27]	Image enhancement using adaptive gamma correction.
[28]	Artifact removal using DullRazor.
[31]	Image enhancement using mean subtraction and standard deviation-based normalization.
[32]	Artifact removal and image enhancement using color constancy with shades of gray.
[33]	Artifact removal and image enhancement using histogram-based preprocessing.
[34]	Artifact removal using a deep learning method.
[36]	Artifact removal using DullRazor.
[37]	Artifact removal using morphological operations and image enhancement with histogram equalization.
[38]	Artifact removal using DullRazor.
[39]	Artifact removal using DullRazor and image enhancement with global-local contrast stretching.
[40]	Artifact removal using median filter and image enhancement with contrast-limited adaptive histogram equalization.
[41]	Artifact removal using Frangi Vesselness filter and image enhancement with contrast-limited adaptive histogram equalization.
[47]	Artifact removal using DullRazor and image enhancement with adaptive histogram equalization.
[48]	Artifact removal using DullRazor with a median filter.
[49]	Image enhancement using adaptive histogram equalization.
[50]	Image enhancement using contrast limited adaptive histogram equalization.
[51]	Image enhancement using contrast limited adaptive histogram equalization.
[52]	Image enhancement using Z-score transformation.

**Table 2 diagnostics-12-00344-t002:** Methods used for segmentation of skin lesions.

Approach	Study	Method	Preprocessing	Dataset	Images
Supervised	[2]	Deep regional CNN and FCM clustering	Yes	ISIC 2016	1279
	[12]	Deep convolutional network	Yes	PH2	200
				ISBI 2017	2750
	[26]	Saliency based	Yes	ISIC 2017	2150
	[30]	FCN based	Yes	ISIC 2016	1279
				PH2	200
	[31]	Recurrent, residual convolutional neural network	Yes	ISIC 2017	2000
	[32]	CNN based ensemble	Yes	ISIC 2018	2594
	[33]	Hybrid learning, particle swarm optimization	Yes	ISIC 2017	550
	[34]	Semantic segmentation based on u-Net	Yes	ISIC 2018	2594
	[35]	R2AU-Net	No	ISIC 2018	2594
	[59]	Deep convolutional encoder-decoder	No	PH2	200
Unsupervised	[8]	Statistical region merging	Yes	Private	90
	[9]	Thresholding	Yes	ISIC 2017	600
	[10]	Stochastic region merging	No	PH2	200
				ISIC 2018 validation	100
				ISIC 2018 test	1000
	[17]	Thresholding	Yes	Private dataset	85
	[19]	Saliency and thresholding	Yes	PH2	18
				ISBI 2016	13
	[20]	K-means clustering	Yes	Dermatology information system andDermQuest	50
	[21]	K-means clustering	Yes	Atlas dermoscopy dataset	80
	[22]	Fuzzy C-Means clustering	Yes	UMCG	170
	[23]	Data clustering	Yes	PH2	200
				ISIC (2016–2019)	5400
	[24]	Saliency	No	EDRA	566
				PH2	200
				ISBI 2016	900
	[25]	Saliency	Yes	PH2	200
				ISBI 2016	900
	[27]	Saliency	Yes	PH2	50
				ISBI 2016	70
	[28]	Saliency and thresholding	Yes	PH2	200
				ISBI 2016	900
	[29]	Multi scale superpixel segmentation	No	PH2	200
				ISBI 2016	900
	[37]	Thresholding and edge detection	Yes	PH2	200
	[38]	Saliency	Yes	PH2	200
	[39]	Region merging	Yes	PH2	200
				ISIC 2017	900
	[40]	Thresholding	Yes	PH2 Mednode DermNet	992
	[50]	Thresholding and GraphCut	Yes	DSSA	294
	[52]	Partially homomorphic POB number system	Yes	PH2	200
				ISBI 2016	1279
				ISBI 2017	2600
	[60]	Superpixel clustering and thresholding	No	PH2	200
	Ours	Saliency-based color histogram clustering with thresholding	No	PH2	200
				ISIC 2018	2594
				HAM10000	10,015

**Table 3 diagnostics-12-00344-t003:** Description of heterogeneous properties inherent in dermoscopic images.

Image Property	Property Description
1	Images with irregular skin lesion shape
2	A large skin lesion that connects multiple image boundaries
3	Skin lesion with low contrast to the surrounding skin
4	Skin lesion with color chart artifact
5	Skin lesion with hair artifact
6	Skin lesion with marker ink artifact
7	Skin lesion with ruler artifact
8	Skin lesion with blood vessel artifact
9	Skin lesion with gel bubble artifact
10	Image with vignette noise artifact
11	Skin lesion with multiple artifacts
12	Skin lesion with multiple shades of color intensity
13	Small skin lesion

**Table 4 diagnostics-12-00344-t004:** Paired samples test for preprocessing effect using PH2 dataset.

Variable			Mean	Std. Err.	Std. dev.	[95% CI]	t-Value	df	Sig ^a^
Accuracy	Pair 1	Without preprocessing	0.921	0.009	0.127	0.903–0.939	2.043	199	0.042
		With artifact removal	0.919	0.009	0.130	0.901–0.938			
	Pair 2	Without preprocessing	0.921	0.009	0.127	0.903–0.939	−3.9213	199	0.000
		With image enhancement	0.933	0.008	0.118	0.917–0.950			
Dice	Pair 3	Without preprocessing	0.893	0.007	0.105	0.878–0.908	0.953	199	0.342
		With artifact removal	0.891	0.008	0.106	0.876–0.906			
	Pair 4	Without preprocessing	0.893	0.007	0.105	0.878–0.908	−4.0814	199	0.000
		With image enhancement	0.909	0.007	0.942	0.896–0.922			

Std. Err. = standard error; Std dev. = standard deviation; Sig = significance; ^a^ (2-tailed); CI = confidence interval; df = degrees of freedom.

**Table 5 diagnostics-12-00344-t005:** Paired samples test for preprocessing effect using ISIC 2018 dataset.

Variable			Mean	Std. Err.	Std. dev.	[95% CI]	t-value	df	Sig ^a^
Accuracy	Pair 1	Without preprocessing	0.923	0.002	0.113	0.918–0.927	1.777	2593	0.076
		With artifact removal	0.921	0.002	0.114	0.917–0.926			
	Pair 2	Without preprocessing	0.923	0.002	0.113	0.918–0.927	−0.096	2593	0.924
		With image enhancement	0.923	0.002	0.112	0.918–0.927			
Dice	Pair 3	Without preprocessing	0.813	0.004	0.179	0.806–0.820	0.651	2593	0.515
		With artifact removal	0.812	0.003	0.178	0.806–0.819			
	Pair 4	Without preprocessing	0.813	0.004	0.179	0.806–0.820	4.953	2593	0.000
		With image enhancement	0.803	0.004	0.189	0.795–0.810			

Std. Err. = standard error; Std dev. = standard deviation; Sig = significance; ^a^ (2-tailed); CI = confidence interval; df = degrees of freedom.

**Table 6 diagnostics-12-00344-t006:** Paired samples test for preprocessing effect using HAM10000 dataset.

Variable			Mean	Std. Err.	Std. dev.	[95% CI]	t-value	df	Sig ^a^
Accuracy	Pair 1	Without preprocessing	0.910	0.001	0.105	0.908–0.912	4.765	10,014	0.000
		With artifact removal	0.909	0.001	0.106	0.907–0.911			
	Pair 2	Without preprocessing	0.910	0.001	0.105	0.908–0.912	−0.7440	10,014	0.000
		With image enhancement	0.914	0.001	0.103	0.912–0.916			
Dice	Pair 3	Without preprocessing	0.824	0.002	0.153	0.821–0.827	6.339	10,014	0.000
		With artifact removal	0.821	0.002	0.157	0.818–0.824			
	Pair 4	Without preprocessing	0.824	0.002	0.153	0.821–0.827	3.801	10,014	0.000
		Without preprocessing	0.820	0.002	0.169	0.817–0.823			

Std. Err. = standard error; Std dev. = standard deviation; Sig = significance; ^a^ (2-tailed); CI = confidence interval; df = degrees of freedom.

**Table 7 diagnostics-12-00344-t007:** Performance comparison of CHC-Otsu algorithm against leading methods on 200 images from PH2 dataset.

Method	Accuracy	Sensitivity	Specificity	Dice
SSLS [38] ^a^	0.85	0.75	0.98	0.78
ASLM [37] ^a^	0.90	0.80	0.97	0.83
[59] ^b^	0.89	0.92	0.87	0.87
[52] ^a^	0.86	0.83	0.92	0.88
[60] ^a^	0.90	0.91	0.89	0.89
SDI+ [9] ^a^	0.91	0.92	0.90	0.85
[10] ^a^	0.92	0.84	0.96	0.90
SPCA [36] ^b^	0.87	0.73	0.95	0.80
YOLO [12] ^b^	0.93	0.84	0.94	0.88
CHC-Otsu ^a^	0.92	0.85	0.98	0.89

^a^ unsupervised method; ^b^ supervised method.

**Table 8 diagnostics-12-00344-t008:** Performance comparison of CHC-Otsu algorithm against leading methods on 2594 images from ISIC 2018 dataset.

Methods	Accuracy	Sensitivity	Specificity	Dice
SDI+ [9] ^a^	0.87	0.87	0.89	0.75
SPCA [36] ^b^	0.84	0.59	0.92	0.62
[34] ^b^	0.93	0.87	0.97	0.87
RU-Net [31] ^b^	0.88	0.79	0.93	0.68
R2U-Net [31] ^b^	0.90	0.73	0.97	0.69
Attention ResU-Net [35] ^b^	0.92	0.84	0.95	0.85
R2AU-Net [35] ^b^	0.93	0.82	0.97	0.87
CHC-Otsu ^a^	0.92	0.78	0.99	0.81

^a^ unsupervised method; ^b^ supervised method.

**Table 9 diagnostics-12-00344-t009:** Performance comparison of CHC-Otsu algorithm against leading methods on 10,015 images from the HAM10000 dataset.

Methods	Accuracy	Sensitivity	Specificity	Dice
SDI+ [9] ^a^	0.90	0.88	0.94	0.83
SPCA [36] ^b^	0.85	0.62	0.96	0.70
CHC-Otsu ^a^	0.91	0.77	0.99	0.82

^a^ unsupervised method; ^b^ supervised method.

## Data Availability

The challenge datasets are available at https://challenge.isic-archive.com/data (accessed on 4 June 2021). The PH2 dataset is available at https://www.fc.up.pt/addi/ph2%20database.html (accessed on 4 June 2021). The HAM10000 dataset is available at https://doi.org/10.7910/DVN/DBW86T (accessed on 20 June 2021).

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
