# Peer review of "Preprocessing Effects on Performance of Skin Lesion Saliency Segmentation"

_diagnostics, 2022, doi:10.3390/diagnostics12020344_

Round 1

Reviewer 1 Report

About the novelty of the content, the authors are not providing a novel idea, but are doing a comparison study about the effect of some preprocessing techniques over the results of segmentation.
However, my presented comments were:

An excellent paper, just some minor issues that need to be addressed:

  • The table of Algorithm 1 CHC-Otsu needs to be well formating to be clear for the reader.
  • Some size adjustments for figures 4 and 5.
  • English editing is necessary (minor mistakes).

Author Response

Comments from Reviewer 1

“About the novelty of the content, the authors are not providing a novel idea, but are doing a comparison study about the effect of some pre-processing techniques over the results of segmentation. However, my presented comments were:

An excellent paper, just some minor issues that need to be addressed:

 Response:

            We really appreciate the good comments from the reviewer.

 “The table of Algorithm 1 CHC-Otsu needs to be well formating to be clear for the reader”

 Response:

Table of Algorithm 1 has been formatted as per the requirement.

“Some size adjustments for figures 4 and 5.”

 Response:

 Figures 4 and 5  have been updated.

“English editing is necessary (minor mistakes).”

Response:

               All identified errors have been adequately corrected and English editing has been thoroughly carried out to fix minor mistakes.

Reviewer 2 Report

This study investigated the effect of preprocessing on the performance of the CHC- Otsu algorithm for saliency segmentation of skin lesions. The paper is really interesting and well document therefore I reccomend the publication

As alraeady reported the manuscript is really interesting considering its focus about melanoma diagnosis and relationship with image segmentation. Many artifacts could be reported durigng the process of segmentation and make preprocessing necessary. 
The collaboration of color histogram clustering with Otsu thresholding are the suggested segmentation methods in this paper. Authors concluded that with this method there is no significant difference in employing a preprocessing method to prepare dermoscopic images for lesion segmentation

Author Response

Comments from Reviewer 2

 “This study investigated the effect of pre-processing on the performance of the CHC- Otsu algorithm for saliency segmentation of skin lesions. The paper is really interesting and well document therefore I recommend the publication

As already reported the manuscript is really interesting considering its focus about melanoma diagnosis and relationship with image segmentation. Many artifacts could be reported during the process of segmentation and make pre-processing necessary. 

The collaboration of color histogram clustering with Otsu thresholding are the suggested segmentation methods in this paper.

Authors concluded that with this method there is no significant difference in employing a pre-processing method to prepare dermoscopic images for lesion segmentation”

Response:

We sincerely appreciate the good comments and thank you for the words of encouragement.